



# BEATBOX: Background Error Analysis Testbed with Box Models

Christoph Knote[1], Jérôme Barré[2], Max Eckl[1,*]

[1]Meteorological Institute, LMU, Munich, 80333, Germany
[2]Atmospheric Chemistry Observations and Modeling, NCAR, Boulder (CO), 80302, USA
*now at: Institute of Atmospheric Physics, DLR, Oberpfaffenhofen, 82234, Germany

*Correspondence to*: Christoph Knote (christoph.knote@physik.uni-muenchen.de) and Jérôme Barré (barre@ucar.edu)

**Abstract.** The Background Error Analysis Testbed (BEATBOX) is a new data assimilation framework for box models. Based on the BOX Model eXtension (BOXMOX) to the Kinetic Pre-Processor (KPP), this framework allows to conduct performance evaluations of data assimilation experiments, sensitivity analyses and detailed chemical scheme diagnostics

from an Observation Simulation System Experiment (OSSE) point of view. The BEATBOX framework incorporates an observation simulator and a data assimilation system with the possibility of choosing ensemble, adjoint or combined sensitivities. A user-friendly, python-based interface allows tuning of many parameters for atmospheric chemistry and data assimilation research as well as for educational purposes, e.g. observations error, model covariances, ensemble size, perturbation distribution on initial conditions, and so on. In this work, the testbed is described and two case studies are

presented to illustrate: the design of a typical OSSE experiment, data assimilation experiments, a sensitivity analysis and a method for diagnosing model errors. BEATBOX is released as an open source tool for the atmospheric chemistry and data assimilation communities.

## 1 Introduction

Current regional and global models of the composition of Earth's atmosphere exhibit a high level of complexity due to the

combination of chemical and meteorological processes such as transport, thermodynamics, radiation or precipitation. But 'just' gas phase chemistry itself already has a considerable amount of complexity. The number of variables (chemical compounds) and equations (chemical reactions) in current model representations of tropospheric chemistry ('mechanisms') can vary by two orders of magnitude (from $10^2$ to $10^4$). Compare, for example, two well-known mechanisms: The Model for Ozone And Related Chemical Tracers, version T1 (MOZART-T1; Emmons et al., 2010; Knote et al., 2014) uses 134 species

and 250 reactions, whereas the Master Chemical Mechanism, version 3.3 (MCMv3.3; Jenkin et al., 2015) employs over 5000 species and over 15 000 reactions. MCMv3.3 is called a 'near-explicit' chemical mechanism, which describes in detail the gas-phase chemical processes involved in the tropospheric degradation of volatile organic compounds (VOC). On the other hand, MOZART-T1 uses a simplified ('lumped') representation of VOCs. This reduction in complexity is advantageous as it reduces the computational demand, but can lead to significant errors in the prediction of atmospheric composition. The

choice of a chemical mechanism is therefore a trade-off: while MCMv3.3 would be desirable due to its fidelity in



representing atmospheric chemistry, it cannot currently be used in large scale 3-D atmospheric simulation due to its computational demand. MOZART-T1 is less accurate, but also much more economic.

Investigating the performance of chemical mechanisms is often done using zero-dimensional (0-D) box models. Numerous studies have used box models to study reactive gas phase chemistry and provide intercomparisons and validation of

mechanisms. Emmerson et al. (2009) provided a comprehensive comparison of the MCM mechanism with six tropospheric chemistry lumped schemes that could be used within chemistry transport models. Archibald et al. (2010) performed an intercomparison of the gas phase mechanism for isoprene degradation in a box model with various mechanisms widely used in 3-D models. More recently, Coates et al. (2015) compared MCM to simplified VOC mechanisms to look at ozone production for a selection of VOCs representative of urban air masses. Knote et al. (2015) conducted box model simulations

using different chemical mechanisms and compared them to each other to understand mechanism-specific biases during a 3-D model intercomparison. Mazzuca et al. (2016) used an observation-constrained box model with a lumped carbon-bond mechanism to study photochemical oxidation and ozone production processes along a research aircraft campaign. Wolfe et al. (2016) presented a tool for 0-D atmospheric modeling that can use different chemical mechanisms and methods of photolysis frequencies calculations.

This non-exhaustive list of recent studies using box models to study tropospheric chemistry shows the importance and need of such tools. Box models are attractive due to their simplicity and low computational cost, but cannot provide a realistic representation of the entire atmosphere because they lack vertical and horizontal diffusion, boundary conditions and numerous other processes that 3-D models take into account. In that regard, box-model studies should not focus on replicating the most accurate predictions but rather aim at gaining significant fundamental and conceptual understanding of a

given system of ordinary differential equations (ODE), in the present case the one that governs tropospheric chemistry.

Another interesting aspect of box-models or reduced complexity models is the applicability to data assimilation research. Low dimension and/or box-models are often used to design new data assimilation algorithms and to prove conceptually the advantage of a given method. In data assimilation theory, use of the Lorentz model and other simple systems is common practice: Sandu et al. (2005) used a box model approach to design an adjoint-based sensitivity analysis for reactive gas phase

tropospheric chemistry. Ott et al. (2004) used a Lorenz-96 model to introduce new formulation of the ensemble Kalman filter approach. Van Leeuwen (2010) also used the Lorenz model to investigate non-linear advanced data assimilation technique such as the particle filter.

Atmospheric composition data assimilation, and more generally inversion, is complex and is computationally demanding. In fact, reactive gas-phase photochemistry is highly non-linear and has to deal with hundreds to thousands of variables. There is

a need for a tool that allows exploring suitable novel data assimilation approaches for atmospheric chemistry, assess uncertainty and errors of a given chemical mechanism and perform chemical sensitivity analysis from a parameter to another. All this should be possible at minimal computational expenses and coding skills requirement. In this paper, we present a new framework based on BOXMOX (BOX MOdel eXtension to KPP; Knote et al., 2015) called BEATBOX (Background Error Analysis Testbed with Box Models) that is able to cycle data assimilation windows, calculate adjoint,



ensemble or even more advanced sensitivity analysis and assess box-model errors and uncertainties. We present in detail the structure of BEATBOX and its algorithms, exemplified through case studies.

## 2 Design of BEATBOX

The Background Error Analysis Testbed with Box Models (BEATBOX) is a suite of tools that allows for a simple and fast
investigation, comparison and evaluation of any system of Ordinary Differential Equations (ODE) evolving over time. By "Background Error" we designate a general term for model error characterization. In this study, BEATBOX is used within the scope of atmospheric chemical mechanisms. Currently, the BEATBOX framework consists of a forecasting tool, the BOX MOdel eXtension (BOXMOX; Knote et al., 2015) to the Kinetic  PreProcessor (KPP; Sandu and Sander, 2006), presented in Sect. 2.1, and a data assimilation tool,  which will be introduced in Sect. 2.2.
The BEATBOX structure is built upon Observing System Simulation Experiments (OSSE; Arnold and Dey, 1986). OSSEs are generally used in the field of numerical weather prediction (e.g. Kuo et al., 1998; Wang et al., 2008; Liu et al., 2009) as well as atmospheric composition and air quality predictions (e.g. Edwards et al., 2009; Claeyman et al., 2011; Barré et al., 2015, 2016). OSSEs allow assessing the benefit of a potential new type of instrument for environmental predictions using a data assimilation system. OSSEs are of crucial importance to define requirements of a given instrument. Also, the model and
data assimilation requirements should be assessed to meet a required predictive capability. The current BEATBOX framework employs a box modeling approach to avoid the space dimension problem - the dimensionality of the problem is reduced to time and variables only. Hence the geometry, radiative specifications, and spatial resolution of a new instrument type are irrelevant. What can be easily explored within BEATBOX is the type of data assimilation method used (e.g. background error covariance calculations, localizations, inflation methods), the revisit time (or temporal sampling) of a
measurement, the variable(s) observed, and straightforward comparisons between sets of ODEs (chemical schemes in our case).

Ultimately an OSSE should be used to highlight model deficiencies and inaccuracies, and provide direct guidance for model improvement. In that context, BEATBOX could be considered as a derived OSSE framework focused on data assimilation technique and model improvement rather than the benefit of new or future type of observations. Starting from a scientific
question or hypothesis to be validated (or rejected) that fits the topics mentioned above, several components are required:

- A nature run (NR) considered as the "true state". The NR supposes to use the best model representation possible considering the state of the art. In this study, we used the Master Chemical Mechanism (MCMv3.3.1) as the NR.

- A control run (CR) as the prior estimated state of the atmosphere. Compared to the NR, a simplified or degraded model should be used for example a set of ODEs that can be implemented in large scale 3-D models. In this study,
we use MOZART-T1.

- An observation simulator that generates synthetic observation by sampling the NR. Observation errors also need to be simulated.




- An assimilation run (AR) that is produced using the data assimilation tool merging the synthetic observation with the CR in order to produce the best estimate possible of the state.
- A suite of diagnostic tools that use NR, CR and AR designed in order to be able point out model and data assimilation technique limitations, ultimately providing a direct feedback for model improvement.

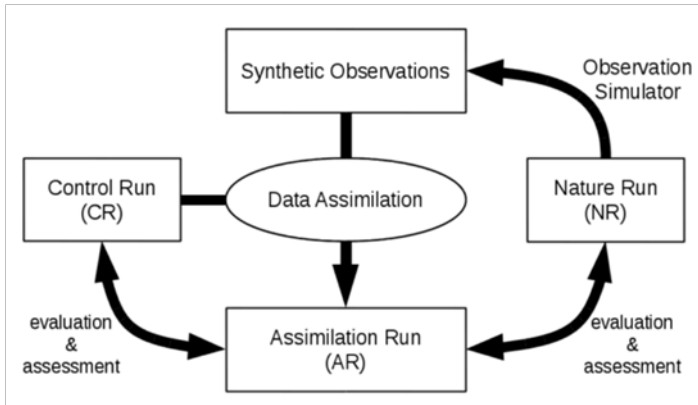

**Figure 1: General flow chart of the BEATBOX system.**

The BEATBOX framework has the capability to loop over several assimilation cycles, also called 'cycling'. Cycling with BEATBOX is schematically displayed in Fig. 2. Every cycle starts with the forecast step. By applying their respective model, the NR, the forecast (F) and the CR are processed from cycle $t-1$ to cycle $t$. Then, $NR_t$ is used to generate synthetic observations through the observation operator $H$ (see Sect. 2.2.2). These observations are assimilated into the forecast $F_t^j$ to produce the analysis $A_t^j$ with, $j$ the data assimilation method of choice using a gain $K^j$ (see Sect. 2.2.3-5). Then, $A_t^j$ will be used to generate new initial conditions $IC_t^j$ to start a new forecast. $CR_t$ serves as a reference to determine the performance of the assimilation method $j$ after $t$ forecast cycles. The forecast $F_t^j$ can be taken to quantify the skill of the assimilation method $j$ at the current cycle $t$. Afterwards, the cycle $t+1$ starts with its forecast step.

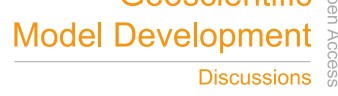

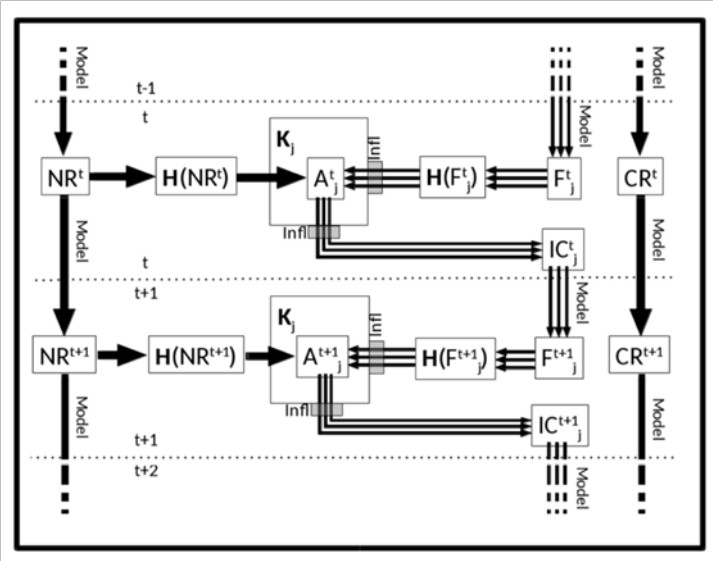

**Figure 2: The cycling sequence with BEATBOX; with the assimilation window t, the assimilation method j, control run CR, nature run NR, observation operator H, gain K, analysis A, initial condition IC, forecast F.**

## 2.1 Forecasting tool

### 2.1.1 The box model BOXMOX

BOXMOX performs box model simulations with different chemical mechanisms using varying sets of input parameters. BOXMOX relies on KPP, a code generator which simplifies the numerical integration of systems of ODEs. The temporal evolution of concentrations of chemical compounds due to photochemistry is a prime example of such a system. KPP takes a predefined set of chemical equations (in our case a chemical mechanism) written in a symbolic, human-readable language

and generates a computer code (FORTRAN, Matlab, C) containing a numerical solver to integrate the system over time. A number of integration methods are available (e.g. Rosenbrock or Runge-Kutta methods). In addition to predicting the evolution of concentrations over time, the resulting solver also delivers the Jacobian and the Hessian matrices of the system. Adjoint models can be generated and tuned (Sandu et al., 2003) and the Jacobians of the adjoint of the model can be obtained (see Sect. 2.2.3). Building upon KPP, BOXMOX provides additional processes typically used in box model studies

(emissions, photolysis, deposition, mixing) and allows for convenient data input. BOXMOX makes simulations of chamber experiments, Lagrangian-type air parcel studies and a description of the chemistry in the atmospheric boundary layer feasible without effort. Input is done via simple text files; initial conditions, photolysis rates, temperature, boundary layer height, de-/entrainment, turbulent mixing, emission, and deposition are possible input parameters. In this work we have extended BOXMOX with an interface written in the Python language (`boxmox` package), to interface more easily with BEATBOX.

BEATBOX uses BOXMOX to rapidly generate a large number of simulations with perturbed input parameters. Temperature, (time-varying) photolysis rates and initial concentrations of each species included in the investigated chemical

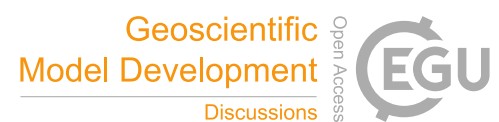



mechanism can be perturbed independently. Ensemble members can be generated by producing normal or log-normal-distributed perturbation factors with possibility to adjust the mean and the standard deviation for each perturbed variable. These perturbation factors are then multiplied by the relevant initial values to produce an ensemble of initial conditions.

In this work, we demonstrate the BEATBOX capabilities using MCMv3.3.1 as NR. Because of its near-explicit representation of atmospheric chemistry, MCMv3.3.1 is a good choice to be seen as the assumed "truth", within the context of an OSSE. The MOZART-T1 chemical mechanism is employed as the simplified/degraded model (CR, AR).

### 2.1.2 Input data generation

BOXMOX comes with a tool to generate input data out of field campaign observations (the `genbox` Python package). Translation to mechanism-specific species naming and lumping is achieved using a translation tool (the `chemspectranslator` package) originally based on the emission database created by Bill Carter (UC Riverside, http://www.cert.ucr.edu/~carter/emitdb). The current system uses data collected in the FRAPPE (Front Range Air Pollution and Photochemistry Experiment) field campaign (`frappedata` package), but could easily be extended to use other, similar datasets from other field campaigns or data sources.

During FRAPPE, a number of flight measurements with remote sensing as well as in-situ devices of numerous quantities have been performed, including concentrations of chemical species, photolysis rates and temperature. In the examples shown we use measurements taken during FRAPPE with the NCAR C130 research aircraft to initialize the box model. Photolysis rates have been measured during FRAPPE and are used in the examples shown here. For other cases where photolysis rates are missing, we provide the ability to use photolysis rates calculated by the Tropospheric Ultraviolet Visible (TUV) radiation model, version 5.1 (Madronich and Flocke, 1997) in BOXMOX using the `tuv` Python package. In this current version of the code only measurements taken during the FRAPPE campaign have been used, but data from other field campaigns can be easily adapted as well with minimal code development.

### 2.2 Data assimilation tool

The `beatboxtestbed` Python package provides a data assimilation tool that sample observations from the NR, with possibility of tuning the observation error parameters. With assimilating observations, different sensitivity analyses can be used, such as adjoint, ensemble or combined sensitivity (in this paper called *hybrid*) are included in this version of BEATBOX (see Sect. 2.2.3-5). For notation purposes, we represent x and y as variables in model or state space and observation space, respectively.

### 2.2.1 The data assimilation problem

Data assimilation combines observations and model information (also called forecast or background) in order to derive an optimal state (analysis) with a reduced error in order to provide the best initial condition for a subsequent forecast. Consider



$n \in \mathbb{N}$ and $p \in \mathbb{N}$ the dimensions of the state (or model) space and the observation space, respectively. Following Nichols (2010) the solution to the data assimilation problem is commonly expressed as follows:

$$\boldsymbol{x}_a = \boldsymbol{x}_b + \boldsymbol{K}(y_o - H(\boldsymbol{x}_b)) \tag{1}$$

where $\boldsymbol{x}_a$ is the analysis state, $\boldsymbol{x}_b$ the background state, $y_o$ the observation, $H$ the observation operator (also known as

forward operator in the variational formalism) and $\boldsymbol{K}$ the gain matrix. The gain matrix $\boldsymbol{K}$ handles the transformation from the observation space to the state (or model) space. Conversely $H$ handles the transformation from model space to observation space. The above equation can also be expressed in the incremental form such as:

$$\Delta \boldsymbol{x} = \boldsymbol{K} \Delta y \tag{2}$$

with $\Delta \boldsymbol{x}$ called the increment and $\Delta y$ called the innovation (or departure). The innovation in the observation space is then

translated to the model space by applying the gain matrix $\boldsymbol{K}$. Different methods exist to estimate the $\boldsymbol{K}$ gain, see Sect. 2.2.3-5. Commonly the gain matrix is given by:

$$\boldsymbol{K} = \boldsymbol{B} H^T (H \boldsymbol{B} H^T + \boldsymbol{R}) \tag{3}$$

where $\boldsymbol{B}$ and $\boldsymbol{R}$ are the error covariance matrices of the background (or model) and observation, respectively. In the BEATBOX framework a single observation ($p = 1$) and only the dimension along the chemical variables is considered.

Hence, $\boldsymbol{R}$ simplifies to a $1 \times 1$ matrix, a scalar $\sigma_o^2$ the observation error variance, $H \boldsymbol{B} H^T$ also simplifies to a scalar $\sigma_b^2$ (see Sect. 2.2.2 below) the background error variance in the observation space. Then $\boldsymbol{B} H^T$ can be seen as $\sigma_b^2 \boldsymbol{s}$ where $\boldsymbol{s}$ can be called here the sensitivity vector. The gain matrix in Eq. (3) then becomes a vector $\boldsymbol{\kappa}$ and can be then reformulated as follows:

$$\boldsymbol{\kappa} = \sigma_b^2 (\sigma_b^2 + \sigma_o^2)^{-1} \boldsymbol{s} \tag{4}$$

**2.2.2 Observation generator – synthetic observations**

Generating observations consists of the following steps:

- Sampling values from the nature run
- Perturbing those values to simulate an observation inaccuracy
- Specifying an observation error value

BEATBOX forecasts have no dimensions in the 3-D atmospheric space (latitude, longitude, altitude). Sampling the NR to simulate observations is straightforward. Conventionally, $H$ is defined as the observation operator and handles the transformation of information defined in the model space to the observation space in order to compare model and observation quantities. Then, the $H$ operator can be expressed as $H^T = [0, 0, \ldots, 1, \ldots, 0]$. If no observation error is simulated the observation is defined as a perfect observation and the observation value is $y_o = H(x_{NR})$.

If the observation error needs to be simulated, then $y_o$ is generated by adding a perturbation. In that version of BEATBOX the perturbation is assumed to be a normal distribution. But other probability density functions can be implemented easily (a log-normal perturbation is also implemented in this version). Then,



$$y_o = H(\boldsymbol{x}_{NR}) + \mathcal{N}(M, \Sigma) \tag{5}$$

with $\mathcal{N}(M, \Sigma)$ a normal distribution of mean M and standard deviation $\Sigma$. The latter two quantities can be view as the observation bias or accuracy (M) and precision ($\Sigma$). Finally, an associated observation error $\sigma_o$ is associated with $y_o$ for data assimilation. $\sigma_o$ can account for bias and/or precision, overestimating or underestimating those parameters, depending if the effect of observation error needs to be tested in BEATBOX. In the following case study (see Sect. 3) we assume non-biased observation and non-biased observation error with a Gaussian distribution, leading to $\sigma_o = \Sigma$.

### 2.2.3 Adjoint sensitivity

Adjoint sensitivities can be calculated using the KPP Jacobian matrix of output from the adjoint model at a given time step. In our case, we make the approximation that the change of the state $\boldsymbol{x}$ (that includes the observed variable $y$) at the time step $t$ relative to the change of the observed variable $y$ at a previous time step $t-1$ (i.e. $d\boldsymbol{x}_t/dy_{t-1}$ or $dy_t/dy_{t-1}$) is linear. In that current study, we assume $t-1$ and $t$ at the beginning and the end of the assimilation window. The adjoint sensitivity vector of the state $\boldsymbol{x}$ to a given observed variable $y$ at time $t$ can be computed using the Jacobian vectors as follows:

$$\boldsymbol{s} = \frac{d\boldsymbol{x}_t}{dy_t} = \frac{d\boldsymbol{x}_t}{dy_{t-1}} \cdot \left(\frac{dy_t}{dy_{t-1}}\right)^{-1} \tag{6}$$

The adjoint assimilation method runs a single forecast with the forward model and also runs the adjoint model and computes the analysis combining Eq. (1), (4) and (6). The innovation in observation space $\Delta y$ is calculated and the state $\boldsymbol{x}$ is inferred using the $\boldsymbol{\kappa}$ gain that includes the adjoint sensitivity $\boldsymbol{s}$. In this method, $\sigma_b$ should be determined with ad-hoc information and will not change during the cycling process. However different methods exist in order to scale $\sigma_b$ appropriately between each cycle. In the BEATBOX context this can be further explored using the provided OSSE framework.

### 2.2.4 Ensemble sensitivity

Ensemble methods run a perturbed set (ensemble) of model realizations in parallel and derive model error and sensitivity using the created ensemble. This gives multiple realizations $i$ of the model in the observation space $y_i$ and model space $\boldsymbol{x}_i$. The standard deviation of the ensemble (or the ensemble spread) is used to estimate $\sigma_b$ in the observation space, such as $\sigma_y = \sqrt{E[(y_i - E[y_i])^2]} = \sigma_b$ with $E$ the averaging operator. Similarly, the ensemble spread in model space can be estimated as $\boldsymbol{\sigma}_x = \sqrt{E[(\boldsymbol{x}_i - E[\boldsymbol{x}_i])^2]}$. Then statistical assumptions are made to derive the sensitivity between $y$ and $\boldsymbol{x}$. One of the commonly used methods as described by Anderson (2003) is to draw a linear fit in the least-squares sense between $y$ and $\boldsymbol{x}$ using the ensemble members, such as

$$\boldsymbol{s} = (\textstyle\sum \boldsymbol{x}_i y_i)(\textstyle\sum y_i y_i)^{-1} = \frac{\sigma_{xy}}{\sigma_y^2} = \boldsymbol{r}_{xy} \circ \frac{\sigma_x}{\sigma_y} \tag{7}$$

where $\boldsymbol{r}_{xy}$ and $\boldsymbol{\sigma}_{xy}$ are respectively the correlation coefficient and covariance vectors between $y$ and each chemical variable of $\boldsymbol{x}$ and $\circ$ the Schur product operator. Then, the innovation in the observation space $\Delta y_i$ is calculated for each ensemble




member. The ensemble state vector $\boldsymbol{x}_i$ is inferred using the same $\boldsymbol{k}$ gain that includes the same ensemble sensitivity $\boldsymbol{s}$ for each ensemble member.

### 2.2.5 Hybrid sensitivity and further approaches

In the current version of BEATBOX the hybrid sensitivity is defined as a combination between the two approaches

mentioned above: it uses an adjoint sensitivity on each ensemble member, such as

$$\boldsymbol{s}_i = \frac{dx_{t,i}}{dy_{t,i}} = \frac{dx_{t,i}}{dy_{t-1,i}} \cdot \left(\frac{dy_{t,i}}{dy_{t-1,i}}\right)^{-1} \tag{8}$$

In that sense, each ensemble member has its own adjoint sensitivity calculation $\boldsymbol{s}_i$ and its own innovation in observation space $\Delta y_i$. This results to independent or different inference on the state vector for each ensemble member $\boldsymbol{x}_i$. This method is just an example of what can be explored with the BEATBOX system. Because of the simplicity of the code and the low

dimensionality of the problem, advanced techniques can be easily implemented and analyzed. Other hybrid methods and filters, such as a polynomial filter or particle filters, and their benefits for highly non-linear systems, can be explored with ease with the proposed framework, as well.

### 2.2.6 Inflation

For ensemble methods, in order to avoid the filter divergence problem (Fitzgerald, 1971), inflation algorithms are needed. In

the BEATBOX framework we included the inflation method as proposed by Anderson (2007), such as

$$\boldsymbol{x}_i^{infl} = \sqrt{\lambda}(\boldsymbol{x}_i - E[\boldsymbol{x}_i]) + E[\boldsymbol{x}_i] \tag{9}$$

with $\lambda$ called the covariance inflation factor that determines by how much the ensemble members $\boldsymbol{x}_i$ are spread out from the mean $E[\boldsymbol{x}]$. Many different methods exist to estimate $\lambda$; it can be chosen constant over time or adaptive given the ensemble spread in observation space $\sigma_b$, observation error $\sigma_o$ and the innovation norm from the ensemble mean $\theta =|E[y_i]$- $y_o|$. In the

version of BEATBOX presented here we calculate $\lambda$ as

$$\lambda = \frac{\theta^2 - \sigma_o^2}{\sigma_b^2} \tag{10}$$

If $\lambda$ is found smaller than one, the ensemble spread is assumed to be large enough and no inflation is calculated. Straightforward computation of $\lambda$ as above assumes linearity between observation space and model space which is true in the current BEATBOX framework. More advanced ways to compute $\lambda$ as described in Anderson (2007) appendix A have been

tested and implemented in BEATBOX and can be used as well. Finally, to conserve the positive definite nature of the ensembles and also prevent clamping ensemble members that would be inflated to negative values we reduce the inflation factor iteratively on every value of the state such as:

$$\lambda \leq \left(\frac{E[\boldsymbol{x}_i]}{\boldsymbol{x}_i^{infl} - E[\boldsymbol{x}_i]}\right)^2 \tag{11}$$



This is one of several methods to conserve the physical aspects of the ensemble: it might under-disperse the ensemble in some cases of low concentrations but assures the inflation is kept to physical values. In the BEATBOX framework, users can easily implement and explore new inflation methods for non-linear, definite positive and perturbation of highly-sensitive systems, as in the case for reactive gas-phase chemistry.

### 2.2.7 Localization

One should consider to what extent the sensitivities should be relied on. Localization algorithms try to limit the impact on the analysis of errors in the sample covariance between observations and model state variables (Mitchell and Houtekamer, 2000). Depending on the ensemble size or the mathematical assumptions to compute a sensitivity, a localization function $C$ should be used to tell the data assimilation algorithm where to apply the sensitivity $s$, such as:

$$s_{loc} = s \circ C \tag{12}$$

In the current BEATBOX framework, it is possible to specify which species should be inferred, e.g. $C^T = [0, 1, \dots, 1, \dots, 0]$.

### 2.3 Flux method for model analysis

A flux-tool has been included in `boxmox` that calculates the production and loss fluxes for a given chemical component. Consider the generalized chemical reactions such as,

$$\sum_i [R]_i \xrightarrow{k} \sum_j [P]_j \tag{13}$$

with $k$ called the rate constant. The chemical flux can be expressed as

$$\frac{d[R]}{dt} = -\frac{d[P]}{dt} = -k \prod_i [R]_i = k \prod_j [P]_j \tag{14}$$

For a given chemical component, a detailed budget of chemical production and loss can be made identifying different chemical reactions. An example of the application of the flux tool is shown in Sect. 3.4.

### 3 Application to an urban pollution case study

In this section, we provide an example case study to showcase the capabilities of the BEATBOX framework.

### 3.1 Control Run and Nature Run

Temperature, concentrations and photolysis rates from the FRAPPE data (see Sect. 2.1.2) are used for initial conditions. In the present simulations, environmental parameters such as temperature and photolysis rates are kept constant. Interactions between the simulated box and the surrounding are neglected, the box model simulation can be seen as a 'chemistry in a jar', similar to a chamber experiment without consideration of wall losses, where only the temporal evolutions due to chemical reactions is allowed.



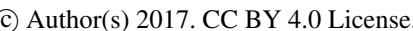


The present case study focuses on an air mass originating from industrial area Commerce City, near Denver, Colorado. Figure 3 shows the estimated temporal evolution over the first 45 hours of simulation for the VOC/NO$_x$ (NO$_x$= NO + NO$_2$) and toluene/benzene-ratio using the MOZART-T1 scheme. The vertical lines suggest the VOC-limited, NO$_x$-limited and the transition region. The measurements show an initially strongly VOC-limited air. The VOC/NO$_x$-ratio of the aging air increases in time. During the first 15 h the simulation shows a strong VOC-limited regime. A transition regime spans from 15 h to approximately 30 h. After the transition period the chemical regime becomes NO$_x$-limited. The toluene/benzene-ratio is used as qualitative measure of photochemical age. Toluene and benzene are considered to have the same sources but toluene is more quickly oxidized than benzene which leads to a decline in the ratio over time.

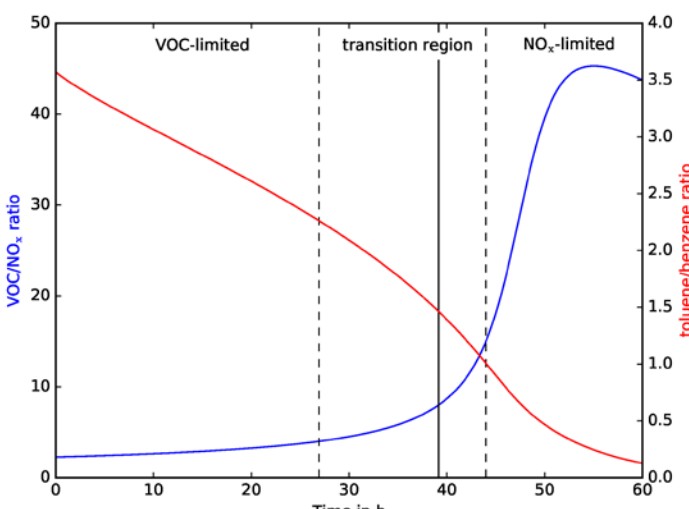

**Figure 3: Temporal evolution of the VOC/NO$_x$ and toluene/benzene ratios over 45h, derived from MOZART-T1 (CR).**

The temporal evolution of some key species in the NR and the CR are displayed in Fig. 4. Nitrogen dioxide (NO$_2$) shows a decrease over time, with a stronger decay in the NR than in the CR. Ozone (O$_3$) production is observed over time with hence a stronger production in the NR than in the CR. The hydroxyl-radical (OH) availability is increasing over time especially between 20 and 35 hours which can be identified as the end of the transition. In the VOC-limited and NO$_x$-limited regimes, the OH increase is slighter. Formaldehyde (CH$_2$O) shows in both NR and CR a decrease followed by an increase and finally a decrease. Those variations illustrate the change of chemical regimes from VOC-limited to transition to NO$_x$-limited. If we compare the NR to the CR the change of chemical regimes is lagged. The change of regimes seems to occur faster in the NR than in the CR suggesting a different reactivity and pathways of oxidation in the NR than in the CR.



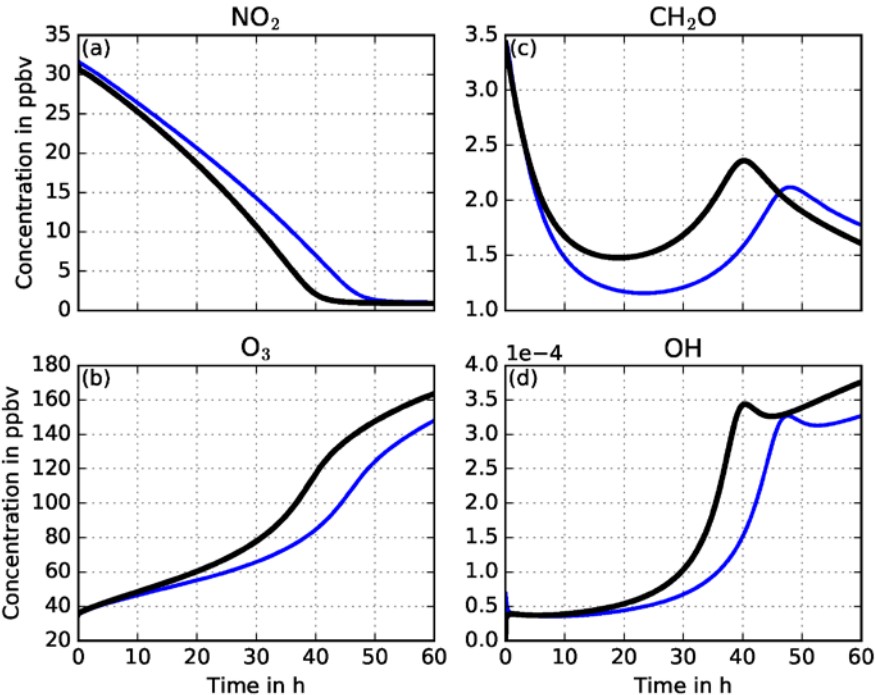

**Figure 4: Time series of concentrations of NO$_2$, O$_3$, HNO$_3$, CH$_2$O, and OH over 45h from the Control Run (MOZART-T1) and from the Nature Run (MCMv3.3.1).**

### 3.2 Assimilation runs

5    We focus on assimilating NO$_2$ and CH$_2$O concentrations in two separate experiments to show the capabilities of the

BEATBOX framework. This is motivated by the fact that NO$_2$ and CH$_2$O play a key role in the atmospheric chemistry

especially for short-lived chemical compounds. Also, NO$_2$ and CH$_2$O are, and will be, observed from space from both low

earth and geostationary orbiters and from in-situ observations. We define a reduced state vector for simplicity of the

demonstration with the species described above: NO$_2$, O$_3$, CH$_2$O, OH.

10   We use an assimilation window of 3 hours. Unbiased observations have been defined and assumed to be at the end of the

assimilation window with an observation error of 0.75ppbv for NO$_2$ and 0.07ppbv for CH$_2$O (corresponding to

approximately 2.5% and 5% of the concentrations at initial time). All the generated observations have well estimated

observation error such as, $\sigma_o = \Sigma$ (see Sect. 2.2.2). The model error in the observation space $\sigma_b$ has to be specified for the

adjoint technique and is set to 2.5% for NO$_2$ and 5% for CH$_2$O of the current forecast concentration (see Sect. 2.2.3). For the

15   ensemble method, the ensemble spread implicitly defines $\sigma_b$ using 50 ensemble members which are generated by perturbing

the initial NO$_2$ concentration by 2.5% for NO$_2$ assimilation and by perturbing the initial CH$_2$O concentration by 5% for





CH$_2$O assimilation (see Sect. 2.2.4). Adaptive inflation is applied on the ensembles at every assimilation window to maintain a realistic ensemble spread in order to weight observation and model appropriately (see Sect. 2.2.6).

### 3.2.1 NO$_2$ assimilation

We assimilate NO$_2$ observations using two different localizations. The first experiment infers only NO$_2$ concentration and the second infers the entire state vector (i.e. NO$_2$, O$_3$, CH$_2$O and OH concentrations). After looking at the evolution of NO$_2$ concentrations (see Fig. 5), all three assimilation methods in both experiments (adjoint, ensemble and hybrid) tend to move the analysis closer toward the observations and hence the NR. Some differences between the three AR can be found. Recalling Sect. 2.2.1 and Sect. 2.2.2, because of the dimensionality of the problem the sensitivity from observation space NO$_2$ to the model variable NO$_2$ is equal to identity, $s = 1$ for every assimilation method. Hence, difference between assimilation runs will only result from differences in $x_b$ and $\sigma_b$ after a subsequent forecast. The ensemble and hybrid methods show a quicker improvement than the single member adjoint method due to the adaptive nature of $\sigma_b$ with the inflation (see Sect. 2.2.6). The single adjoint method keeps $\sigma_b = 2.5\%$ while ensemble and hybrid methods can tune this value with the inflation.

When only NO$_2$ is inferred the other species are modified and this could be called the model response to assimilated changes: NO$_2$ is decreased that is increasing the OH availability for other oxidation pathways. A slight increase of O$_3$ is noted and more significantly for CH$_2$O during the transition region. The ensemble methods have a stronger effect due to the adaptive nature of $\sigma_b$. The adjustment of NO2 concentration towards NO$_2$ observation can be more effective and this can have a stronger effect on the model response.

When the entire state vector is inferred, i.e. no localization, slight additional increases occur on O$_3$, CH$_2$O and OH in the transition region. In general, in the first 10 to 15 hours, when VOC-limited conditions dominate (see Fig. 3) the impact of NO$_2$ assimilation is low. By definition, VOC-limited air is insensitive to changes in NO$_2$, so if the model (CR) predicts VOC-limited conditions. Either adjoint sensitivities or ensemble sensitivity will remain small. After 15 hours of simulation, the chemical regime transitions significantly to NO$_x$-limited with higher sensitivity of the state to changes in NO$_2$. After 30 hours, the chemical regime tends to be NO$_x$-limited, NO$_2$ is not used for secondary production anymore and levels are steady. NO$_2$ assimilation increments are then very small, hence almost no inference on the other state variables is observed. In that case study, inference from NO$_2$ observation on the rest of vector is, in that case, not the major reason for improvement. Model response from NO$_2$ changes is mostly responsible for improving the state. NO$_x$ concentrations drive the chemistry in the transition region and NO$_x$-limited regimes. NO$_x$ chemistry is well known and similarly represented between the NR and the CR. Hence the model response is likely to improve the state and not likely to produce additional errors.




**Figure 5: Evolution of the state vector over 60 hours, including the Nature Run (NR), Control Run (CR) and assimilation runs using adjoint, ensemble and hybrid method. Two experiments with different localizations are displayed, only NO₂ inferred (left) and whole state vector inferred (right). Shaded areas show corresponding ensemble spread for ensemble and hybrid methods.**



### 3.2.2 CH$_2$O assimilation

We repeat the same experiments but assimilating CH$_2$O observations instead. In Fig. 6, the left-hand side plots show the concentration evolution when only CH$_2$O is inferred. In the first 40 hours CH$_2$O is underestimated by the CR mechanism. All three AR show very similar results. During the analysis phases the CH$_2$O concentrations are pushed up towards the NR.

Those abrupt increases are systematically compensated by the chemical mechanism that is willing to come back to chemical equilibrium, i.e. towards the CR concentrations. This makes the CH$_2$O concentration evolution as saw tooth shape. CH$_2$O is a short lived chemical compound (shorter than NO$_2$ in this case study) and it is mainly driven by other chemical species concentrations. Ultimately CH$_2$O concentration in the ARs are slightly risen up after each 3-hour forecast. These changes will very mildly affect the state vector concentrations. The model response of CH$_2$O changes on NO$_2$, O$_3$ and OH is slight

and definitely smaller than the model response on NO$_2$ changes (see Sect. 3.2.1).

When the entire state vector is inferred, i.e. no localization (Fig. 6, right-hand side plots) we observe very different results. The two ARs that are using adjoint sensitivities (hybrid and adjoint) show similar results than the previous experiment, the saw tooth shape is still observed. The analysis shows strong increases on CH$_2$O and the chemical mechanism tries to come back to the CR state. The inference on the other species of the state vector is noticeable, NO$_2$, O$_3$ and OH concentrations are

improved compared to the experiment when only NO$_2$ is inferred.

In the ensemble method where no localization is applied, the improvement on CH$_2$O is of a different nature. The saw-tooth behavior of the AR is not observed anymore and the chemical mechanism now seems to have changed from systematic CH$_2$O loss to CH$_2$O production in the forecast. This will drive the AR to fit better the observations and hence the NR. The inference on other species show significant changes that will in general change the sign of the error and sometimes increases

it; NO$_2$ becomes underestimated in the transition region, O$_3$ is overestimated but then underestimated after 35 hours, and OH is overestimated in the transition region. In the ensemble method for that case study, the computed sensitivities seem significantly different from the adjoint sensitivities. This will allow to change the chemical production/loss rate of CH$_2$O to adjust better the CH$_2$O concentration but at the risk of disturbing the rest of the state vector and increasing errors that might lead to unphysical results. We diagnose the difference between sensitivities from the two experiments presented above in

Sect. 3.3. We also diagnose the chemical behavior change from CH$_2$O assimilation using fluxes in Sect. 3.4.





**Figure 6: Same as Fig. 5 but with CH₂O observations assimilated.**



### 3.3 Sensitivities of the off-diagonal elements of the background error covariance matrix

Figure 7 shows the sensitivities of the unobserved species to a change in $NO_2$ and $CH_2O$ respectively at the end of each 3 h assimilation window with the single member adjoint and the ensemble method (see Sect. 2.2.3 and 2.2.4). To make sensitivities comparable we have normalized (divided) them by the state concentrations. For example, $s(NO_2, O_3)$ the

sensitivity of $NO_2$ changes over $O_3$ will be most likely orders of magnitude larger than $s(NO_2, OH)$ since $O_3$ concentration are orders of magnitude larger than OH concentrations.

For $NO_2$ assimilation both methods deliver similar results. The sensitivities are in general small, not above 60%. Sensitivities are small or negligible in VOC-limited regime but become more significant during the transition and the $NO_x$-limited regimes, mostly after 30 hours. The changes on $NO_2$ from assimilation after 30 hours are rather small and hence the

inference on other species will be small. This then confirms that the most important part of the changes from $NO_2$ assimilation is due to the model response from $NO_2$ changes and secondly due to the data assimilation inference on the other species of the state.

For $CH_2O$ assimilation we saw significant differences in behavior between adjoint and ensemble methods to compute the sensitivities. The adjoint displays rather small sensitivities, peaking at 20% but mostly below 10%. Those sensitives are

observed a during the VOC-limited regime when the chemistry is sensitive to changes from VOC concentrations, and disappear during the transition region and become insignificant in the $NO_x$-limited regime. This explains the small but reasonable impacts form $CH_2O$ changes on the rest of the state. Concerning the ensemble method, the computed sensitivities are much larger and do not decrease after the transition regime. Values switch abruptly from negative to positive and are sometimes above 200%. To understand this suspicious unphysical behavior, we display tracer-tracer relationships during the

assimilation phase of the 9[th] cycle (27 hours), in Fig. 8.

In Sect. 2.2.4, we defined the ensemble method sensitivity computation as a linear-fit to the ensemble distribution between two species. We recall that ensemble methods are not limited to this, but in the current state of the art EnKF methods mostly use this approximation. One can see the limitation to such linear assumption after looking carefully at Fig. 8. For example, on the right-top corner are displayed the prior and posterior states of the $NO_2$-$O_3$ distributions. The ensemble method

provides a very good match to the observation; the ensemble distribution is at the $NO_2$ level of the observation and the ensemble spread fits the observation error (green vertical bars). However, the relationship between $NO_2$-$O_3$ that has formed during the ensemble AR after 9 cycles is strongly curved and non-linear. The ensemble sensitivity is then trying to draw a linear fit on this strongly curved distribution. On this example, moving the ensemble members along the linear-fit is moving the $O_3$ distribution slightly away from the NR. At the same time the adjoint sensitivities are in comparison very weak (the

slope is along the y-axis), slightly improving the state distributions but not very strongly. Finally, the hybrid distribution show lager spread and different distribution shape than the ensemble distribution. The adjoint inference do not strongly changes the rest of the state that maintains the chemical production/loss rates of $CH_2O$ that falls into a temporary attractor; the model wants to go back to the CR concentrations and creates a sawtooth-shaped concentration evolution over time (see

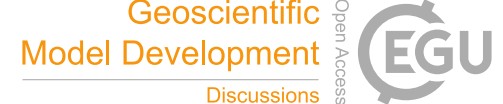

Sect. 3.2.2). After 3 hours, the forecast will be significantly far from the observation and the distribution will be inflated. The ensemble inference strongly changes the rest of the state, that will change the chemical production/loss rates and drive the system out of the attractor. To understand this more clearly, the flux tool is presented in the following section.



5    **Figure 7: Temporal evolution of the sensitivities of the unobserved state species on changes from the observed one at the end of each 3h assimilation for NO$_2$ assimilation (top panel) and CH$_2$O assimilation (bottom panel).**

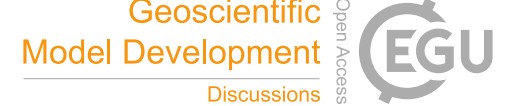



**Figure 8: Tracer-tracer or observation space to state space relationships during the 9th cycle of the CH₂O assimilation without localization experiment.**



### 3.4 Model diagnostics using fluxes

In this last section, we will focus on the $CH_2O$ fluxes (production and loss) over the $9^{th}$ cycle forecast (25 to 27 hours) of the $CH_2O$ assimilation without localization. In Fig. 9 are displayed the individual contributions of production and loss of $CH_2O$ for the NR and one same member of the CR, AR ensemble and AR hybrid. The flux tool isolates a few different reactions

with NR because the chemical scheme is different and more detailed than the CR. In this case study, $CH_3O \rightarrow CH_2O + HO_2$ in the NR is simply an intermediate reaction of $NO + CH_3O_2 \rightarrow CH_2O + NO_2 + HO_2$. Other than this reaction, the isolated reactions are similar.

The NR fluxes to the CR fluxes show the same order of production/loss importance, however the NR fluxes are stronger. Overall the loss terms in the NR are stronger than the CR leading to a net chemical flux that is more at a loss. If we now

compare the CR rates with the AR ensemble, we can see that magnitudes are significantly stronger and the slopes of the rates (i.e. the second derivative of the concentration evolution) is also changed. The production terms are stronger leading to almost no loss net flux. The order of importance of the reactions in the budget is also different; $CH_2O + OH \rightarrow CO + HO_2 + H_2O$ is much more important, probably due to the increase of OH during the AR ensemble (see Sect. 3.2.2). Finally, the AR hybrid fluxes are similar to the CR fluxes with the same order of importance between reactions. Except for the carbon

monoxide (CO) formation from photolysis, $CH_2O \rightarrow CO + H_2$ that is now stronger and that is responsible for the systematic decrease of $CH_2O$ in the AR adjoint and AR hybrid (see Sect. 3.2.2). This increased flux from this reaction explains the saw tooth behavior of $CH_2O$, when no other pathway of loss is possible, i.e. OH is not as strongly changed in AR adjoint and hybrid as in AR ensemble, this way of destroying $CH_2O$ is then increased.

We demonstrate here the usefulness of the flux tool to diagnose the effect of data assimilation on atmospheric chemistry in a

detailed manner. The diagnostic of the flux tool can be used on any species that a chemical scheme contains for any kind of BOXMOX simulation.





**Figure 9: CH₂O fluxes during the 3 hour forecast of the 9ᵗʰ cycle of the CH₂O assimilation with no localization. For the NR (a), one member of the CR (b) one member of the AR ensemble (c) and one member of the AR hybrid (d). Fluxes that are representative of 90% of the budget are displayed.**

## 4 Summary and future options

In this paper, we have presented a new suite of tools for box-models, BEATBOX. The design of BEATBOX is based on an OSSE approach that can integrate simultaneously various chemical schemes to simulate a nature run, control runs and assimilation runs. This framework includes the capability of running assimilation windows of different chemical schemes using a forecasting tool (BOXMOX) and an assimilation tool allowing sensitivity analysis. BEATBOX provide ensemble and adjoint sensitivity analysis that can be combined or modified to explore new inverse or data assimilation methodologies. Additionally, a flux tool is also integrated into the framework to diagnose and assess in detail models runs differences and ultimately use data assimilation to improve the model (chemical scheme) itself and not only the model outputs. The





systematic and detailed assessment of the multivariate data assimilation problem that BEATBOX can tackle important scientific hypotheses at a limited computational cost for future data assimilation configuration on large scale 3D models for the atmospheric chemistry but not only limited to it. Any system of equations can be integrated over time in the current framework.

A typical case study of ozone photochemical production from $NO_x$ is presented to showcase the capability of BEATBOX. Differences between the nature run and the control run are presented followed by a data assimilation experiment of synthetic $NO_2$ and $CH_2O$ observations using adjoint and ensemble sensitivity analysis with different localization parameters. The case studies shown in this paper illustrate the need to understand in detail the effect of data assimilation in as complicated and non-linear a model that the atmospheric chemistry requires. In these case studies, we showcased that BEATBOX is a

powerful and user-friendly tool for:

- Understanding chemical mechanism differences and deficiencies
- Performing chemical sensitivity analysis using ensemble or adjoint methods
- Envisioning and designing new inverse and data assimilation methods for atmospheric chemistry to optimally constrain as much of the chemical state as possible

- Defining requirements for new chemical and data assimilation schemes and ultimately improving them.
- Educational purposes for data assimilation and atmospheric chemistry

BEATBOX will continue to evolve depending on user requirements. For example, emission inversion capability is currently being implemented. Setting any given observation time into the assimilation window will also be possible. Assimilating multiple observations in a given assimilation window will be also implemented using sequential and variational

minimization techniques, as well. Using real observations, i.e. from fields campaigns is also possible with minimal code modifications. Because of the user friendliness, flexibility and the open source nature of most the BOXMOX/BEATBOX, users could also contribute to model development making it a broad atmospheric chemistry community tool.

## 5 Code and/or data availability

All code developments shown here are open source tools released under the GNU General Public License v3. The chemical box model BOXMOX is available for download at https://boxmodeling.meteo.physik.uni-muenchen.de, where it can also be used in an online version. BEATBOX and all other necessary python packages are available through the Python Package Index (https://pypi.python.org). The following packages are available:

- `beatboxtestbed`: the BEATBOX Background Error Analysis Testbed

- `boxmox`: Python interface for the chemical box model BOXMOX
- `genbox`: input data generator for `boxmox`

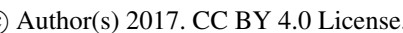



- `frappedata`: FRAPPE campaign dataset for `genbox`
- `tuv`: TUV data connector for `genbox`
- `chemspectranslator`: translator to translate species between chemical mechanisms and observations
- `icartt`: reader/writer for ICARTT files

Full documentation for all Python packages, including examples on how to reproduce the case studies shown in this manuscript can be found at https://boxmodeling.meteo.physik.uni-muenchen.de/documentation .

## 6 Author contributions

CK and JB contributed equally to the manuscript. CK designed the BOXMOX system, wrote the framework around BEATBOX, supervised ME, and contributed to writing the manuscript. JB came up with the idea for BEATBOX and coded

most of the beatbox.py routine, co-supervised ME and wrote most of the manuscript. ME developed parts of the BEATBOX framework during his Masters thesis, prepared and analysed the case studies, and contributed the flux tool.

## 7 Acknowledgments

We thank the many scientists who contributed to the FRAPPE field campaign for providing useful data to initialize the model simulations. Frank Flocke and Rebecca Hornbrook (NCAR) are thanked for creating the 'Commerce City' sample

used in the case studies. We acknowledge Bill Carter (UC Riverside) for his work on the emission database. We acknowledge support from the NASA KORUS-AQ grant NNX16AD96G.

## 8 Competing interests

The authors declare that they have no conflict of interest.

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
