# Peer review of "BEATBOX v1.0: Background Error Analysis Testbed with Box Models"

_Geoscientific Model Development, 2017_

## Editor Comment (EC1) · T. Butler (Editor) · 4 Oct 2017

https://www.geoscientific-model-development.net/about/manuscript_types.html

Model description papers must give the model name and version number (or other unique identifier) in the title. Please make sure that you consider this if you prepare a revised manuscript for publication in GMD.

---

## Short Comment (SC1) · 10 Oct 2017

Christoph,

On the code web page I have noticed that there are various releases of the BEATBOX code. In the code accessibility section could you please specify which release you have used to create the results in the paper? The statement on access through pypi is a bit unclear. I assume you mean that this provides an alternative access to the same code that can be accessed via https://boxmodeling.meteo.physik.uni-muenchen.de as a whole. Can you please clarify? If these packages are needed in addition to BEATBOX you should specify the versions being used in the paper.

GMD is also encouraging authors to provide a persistent access to the released source

code as supplement or through the use of a DOI. At this point this is not compulsory but suggested where appropriate.

All the best Lutz Gross GMD Executive Editor

---

## Author Comment (AC1) · 11 Oct 2017

Dear Lutz,

there is only one release of the BEATBOX code, which is the `beatboxtestbed` Python package currently hosted on PyPi.

I agree that the installation procedure is somewhat confusing at the moment, due to the following reasons:

- We tried to be good Python citizens, hence we published all Python packages we created for the BEATBOX framework on PyPi.

- PyPi hosts packages, but not documentation, which we hence put on our website

at https://boxmodeling.meteo.physik.uni-muenchen.de/documentation/.

- BOXMOX, the underlying chemical box model is not a Python package, but a standalone Linux executable. It is hence not distributed through PyPi, but directly through our website at https://boxmodeling.meteo.physik.uni-muenchen.de/downloads.html

- A Python wrapper package for BOXMOX (aptly named `boxmox`) was created to use BOXMOX from within Python

The BEATBOX framework therefore consists of the BOXMOX chemical box model, a Linux executable to be installed from our website, and a number of Python packages to be installed from PyPi (using `pip`, the de factor standard to install Python packages).

To install the full framework:

1. Install BOXMOX from our website

2. Install the python packages using pip

*Given that you have a reasonably recent Python 2.x version, with numpy and matplotlib packages installed, the second step should boil down to a single*

```
pip install beatboxtestbed
```

*command.*

I hope this clarifies the current code distribution setup, please let me know if further problems arise. In the meantime we will work on improving the situation and explore the best way to provide DOI-ed persistent source code access for the final release, given our manuscript is accepted for publication in GMD.

Best regards, Christoph

---

## Referee Comment (RC1) · Anonymous Referee #1 · 3 Nov 2017

**Review**

This paper describes the data assimilation tool BEATBOX. This open source tool should be very useful to the scientific community working with data assimilation in the area of atmospheric chemistry, both for research and teaching. The description of the tool is comprehensive and the examples illustrative. Thus, this paper will be of great interest to the scientific community.

The paper is suitable for publication once the authors address the comments below. They mainly concern providing further important details about OSSEs, and clarification of points made. The style of the language, including ensuring that the English is clear, is also a concern that needs addressing.

L. 21: Avoid subjective terms like "interesting".

L. 23: Do you mean Lorenz?

L. 28-29: Do you need "In fact"? Avoid needless words. Do this elsewhere in the paper, e.g., consider omitting the form "in order". Consider omitting "suspicious" in L. 19 of P. 17.

P. 3

L. 2: Indicate what you will do in each section of the paper.

L. 14: Perhaps comment that the space agencies, e.g., ESA, now support the use of OSSEs to inform on the performance of proposed missions. The authors could refer to the concept of scientific readiness level.

L. 22: Somewhere in this section, the authors should mention issues with OSSEs to take into account in their design: the cost; the "incest" or twin problem when the models producing the Nature Run and performing the assimilation experiments are the same; interpretation of results.

P.7

Eq. (3): I think a superscript "-1" is missing after the brackets.

P. 9

L. 26: I am not sure what you mean by clamping. Should it be "clumping"?

P. 10

L. 9: Do not anthropomorphize the data assimilation system. I suggest you use a word other than "tell". See also P. 15, L. 5; P. 17, L. 27. There are more instances.

P. 11
L. 14: What transition?

P. 14

Fig. 5: I suggest that the caption includes the description of the line styles. Do this also for similar figures.

P. 15

L. 2: Which same experiments? Identify them here.

P. 17

L. 30-31: Rephrase and correct typos (this is an example of what to avoid). Check carefully the English language throughout the paper.

P. 20

L. 13: I suggest you replace "probably" with "likely".

P. 24

L. 33: The Lahoz et al. reference is not in the main text. Please address.

---

## Referee Comment (RC2) · Anonymous Referee #2 · 21 Nov 2017

This paper describes the BEATBOX framework, which enables users to perform and asses several data assimilation techniques using the BOXMOX model. The tool is open-source and should be of high interest to the atmospheric chemistry community. The online examples provide appropriate guidance to reproduce the results presented here.

I have only minor concerns with the manuscript concerning clarity. There are a few sections that are difficult to follow, and could be corrected by including more details, simplifying sentences, or fixing typos. I recommend publication after addressing the points below.

Page 1, line 8: should be "allows users to conduct"

Is Figure 1 referenced anywhere?

Page 5, line 17: Mention that BOXMOX is a standalone Linux executable (i.e. not written in python).

Page 6, section 2.1.2: The last sentences of each paragraph in this section are essentially the same. Include it only once.

Page 7: line 14: A single observation (p=1) means a single observation in space, but there can be multiple observations in time. Somehow clarify the time dimension here.

Page 8, line 2: should be "can be viewed"

Page 11, line 1: Please provide more information on the setup. What meteorological parameters are varied with time? What is meant by VOC here? All measured VOCs and their oxidation products?

Page 11 line 3: 'VOC-limited, NOX-limited, and transition region" . . . Indicate that these refer to ozone production. How was the placement of the vertical lines determined?

Page 11 line 15: replace "slighter" with "smaller".

Figure 4, legends are needed.

Page 13 line 22: remove period before "either"

Page 13 line 24: what is "secondary" production?

Page 13 line 26: remove one instance of "that case"

Page 15 line 5: replace "is willing to come back to" with "returns"

Page 15 line 6: replace "as" with "a"

Figure 7, bottom panel: are the colorbars saturated more often than not? If so, adjust the colorbar scaling.

Page 15 line 17: "form" should be "from"

Page 17 line 22: Simplify sentence starting with "We recall that"... to read "Most state-of-the-art EnKF methods use this approximation"

Page 17 line 31: Fix "The adjoint inference do not strongly changes"

Page 17 line 33: Change the model "wants to go back to" to "returns"

Page 20 line 9: Replace "more at a loss" with "negative".

Page 20 line 10, Simplify the sentence starting with "If we now...", to read "The CR rates are significantly faster than the AR ensemble rates, and the slopes of the rates (i.e. the second derivative of the concentration evolution) also differ".

Figure 9: These are difficult to compare with the different y axis. Consider an additional plot that contains net production and net loss for each run, and including these all on the same axis (without individual reactions).

---

## Author Comment (AC2) · 12 Dec 2017

*Review*
*This paper describes the data assimilation tool BEATBOX. This open source tool should be very useful to the scientific community working with data assimilation in the area of atmospheric chemistry, both for research and teaching. The description of the tool is comprehensive and the examples illustrative. Thus, this paper will be of great interest to the scientific community.*
*The paper is suitable for publication once the authors address the comments below. They mainly concern providing further important details about OSSEs, and clarification of points made. The style of the language, including ensuring that the English is clear, is also a concern that needs addressing.*

We thank the reviewer for his/her positive review and helpful comments. Please find our responses to the comments below.

*P. 2*
*L. 21: Avoid subjective terms like "interesting".*
We have updated the manuscript on several occasions to avoid subjective terms.

*L. 23: Do you mean Lorenz?*
Yes, we do. Thanks for spotting this typo. Fixed.

*L. 28-29: Do you need "In fact"? Avoid needless words. Do this elsewhere in the paper, e.g., consider omitting the form "in order". Consider omitting "suspicious" in L. 19 of P. 17.*
Removed.

*P. 3*
*L. 2: Indicate what you will do in each section of the paper.*
The last sentence of the introduction now reads: "In section 2 we present in detail the structure of BEATBOX and its algorithms, exemplified through case studies which we discuss in section 3."

*L. 14: Perhaps comment that the space agencies, e.g., ESA, now support the use of OSSEs to inform on the performance of proposed missions. The authors could refer to the concept of scientific readiness level.*
We added the following sentence:
[...] OSSEs allow assessing the benefit of a potential new type of instrument for environmental predictions using a data assimilation system and are of crucial importance to define requirements of a given instrument. Space agencies such as the National Aeronautics and Space Agency (NASA) or European Space Agency (ESA) hence support OSSEs as tools to proof scientific readiness levels for proposed space missions. Also, the model  [...]

*L. 22: Somewhere in this section, the authors should mention issues with OSSEs to take into account in their design: the cost; the "incest" or twin problem when the models producing the Nature Run and performing the assimilation experiments are the same; interpretation of results.*
We clarify the text by adding the following sentences: [...] sets of ODEs (chemical schemes in our case).
A number of issues regarding the OSSE technique should be mentioned as well. Performing an OSSE could be costly in terms of setup and design as well as computationally. Numerical integration of the most state of the art representation of the earth system for sampling observations and benchmarking with could be intensively costly and requires highly skilled staff and extensive collaboration between research entities. Approximations are often required to make experiments possible (e.g. the "identical twin" problem ) necessitating careful diagnosis of the results and could limit scientific conclusions. Ultimately an OSSE should be used to highlight model deficiencies and inaccuracies [...]

*P.7*
*Eq. (3): I think a superscript "-1" is missing after the brackets.*
Corrected.

*P. 9*
*L. 26: I am not sure what you mean by clamping. Should it be "clumping"?*
No, it meant forcing these members to zero instead of becoming negative. The sentence has been revised and now reads: "[...] Finally, to conserve the positive definite nature of the ensembles and also prevent forcing ensemble members to zero that would otherwise be inflated to negative values we reduce the inflation factor iteratively on every value of the state such as: [...]"

*P. 10*
*L. 9: Do not anthropomorphize the data assimilation system. I suggest you use a word other than "tell". See also P. 15, L. 5; P. 17, L. 27. There are more instances.*
We have gone through the text and corrected a number of instances.

*P. 11*
*L. 14: What transition? P. 14*
From the VOC- to the NOx-limited regime. We have updated the text accordingly.

*Fig. 5: I suggest that the caption includes the description of the line styles. Do this also for similar figures.*
We have amended the figure captions to include the line styles shown.

*P. 15*
*L. 2: Which same experiments? Identify them here.*
The sentence has been rephrased and now reads: "[...] We repeat the experiments presented in 3.2.1, but assimilate CH2O observations instead of NO2. [...]"

*P. 17*

*L. 30-31: Rephrase and correct typos (this is an example of what to avoid). Check carefully the English language throughout the paper.*

We have again gone through the manuscript to correct typos. As none of us is an English language native and Copernicus offers English copy-editing upon publication we defer the final language check to their expertise.

*P. 20*

*L. 13: I suggest you replace "probably" with "likely".*

Done.

*P. 24*

*L. 33: The Lahoz et al. reference is not in the main text. Please address.*

Referenced now in section 2.2.1.

---

## Author Comment (AC3) · 12 Dec 2017

Dear editor,

the title of the manuscript has been amended to follow GMD guidelines and now reads:

BEATBOX v1.0: Background Error Analysis Testbed with Box Models

Best regards,

Christoph Knote on behalf of all co-authors

---

## Author Comment (AC4) · 12 Dec 2017

*This paper describes the BEATBOX framework, which enables users to perform and asses several data assimilation techniques using the BOXMOX model. The tool is open-source and should be of high interest to the atmospheric chemistry community. The online examples provide appropriate guidance to reproduce the results presented here.*
*I have only minor concerns with the manuscript concerning clarity. There are a few sections that are difficult to follow, and could be corrected by including more details, simplifying sentences, or fixing typos. I recommend publication after addressing the points below.*

We thank the reviewer for his/her positive review and helpful comments. Please find our responses to the comments below.

*Page 1, line 8: should be "allows users to conduct"*
Fixed.

*Is Figure 1 referenced anywhere?*
It was not, thanks! Fixed, now referenced in the section just above it.

*Page 5, line 17: Mention that BOXMOX is a standalone Linux executable (i.e. not written in python).*
Sentence added: "[...] are possible input parameters. BOXMOX is a standalone C / Fortran program running on Linux or Mac OS X. In this work we have extended BOXMOX [...]"

*Page 6, section 2.1.2: The last sentences of each paragraph in this section are essentially the same. Include it only once.*
We have remove the first occurence.

*Page 7: line 14: A single observation (p=1) means a single observation in space, but there can be multiple observations in time. Somehow clarify the time dimension here.*
Here p=1 means a single observation in the assimilation window (1h), which means a single observation in space and time allotted within the window. We clarify the sentence as follow:
"[...] In the BEATBOX framework a single observation in a given assimilation window ($p$ = 1) and only the dimension along the chemical variables is considered [...]"

*Page 8, line 2: should be "can be viewed"*
Fixed.

*Page 11, line 1: Please provide more information on the setup. What meteorological parameters are varied with time? What is meant by VOC here? All measured VOCs and their oxidation products?*
We have adapted sentence 2 which now reads: "[...] In the present simulations, all environmental parameters such as temperature and photolysis rates are kept constant. [...]" All

meteorological parameters are held constant. A sentence has been added at the end of the first paragraph to clarify what is initialized:
"[...] due to chemical reactions is allowed. Initial conditions for primary VOCs and inorganic compounds are provided using the FRAPPE observations. [...]"

*Page 11 line 3: 'VOC-limited, NOX-limited, and transition region" . . . Indicate that these refer to ozone production. How was the placement of the vertical lines determined?*
The paragraph has been amended to better explain why this was done: "[...] The vertical lines suggest the VOC-limited (VOC/NOx ratio <= 4), NOx-limited (VOC/NOx ratio > 15) and the transition region (4 < VOC/NOx ratio <= 15), to show that the simulation transitions through different O3 production regimes with possibly very different relevant chemical pathways. The measurements show an initially strongly VOC-limited air indicative of an urban air mass. The VOC/NOx-ratio of the aging air increases in time. During the first 15 h the simulation shows a strong VOC-limited regime. A transition regime spans from 15 h to approximately 30 h. After the transition period the chemical regime becomes NOx-limited, representative of more rural / background conditions. The toluene/benzene-ratio is used as qualitative measure of photochemical age. Toluene and benzene are considered to have the same sources but toluene is more quickly oxidized than benzene which leads to a decline in the ratio over time. [...]"

*Page 11 line 15: replace "slighter" with "smaller".*
Done.

*Figure 4, legends are needed.*
Added:

[Figure]

*Page 13 line 22: remove period before "either"*
Done.

*Page 13 line 24: what is "secondary" production?*
This sentence was unclear and has been rephrased: "[...] After 40 hours, NO2 concentrations drop to very low values, NO2 assimilation increments are very small, hence almost no inference on the other state variables is observed. [...]"

*Page 13 line 26: remove one instance of "that case"*
Done.

*Page 15 line 5: replace "is willing to come back to" with "returns"*
Changed, see reviewer 1.

*Page 15 line 6: replace "as" with "a"*
Fixed.

*Figure 7, bottom panel: are the colorbars saturated more often than not? If so, adjust the colorbar scaling.*
We have extended the color bar range, but had to find a balance with visibility of changes in the top panel, as we wanted to keep the color bar range identical for all plots to aid comparability.

[Figure]

*Page 15 line 17: "form" should be "from"*
Thanks for spotting this one!

*Page 17 line 22: Simplify sentence starting with "We recall that". . . to read "Most state-of-the-art EnKF methods use this approximation"*
Done.

*Page 17 line 31: Fix "The adjoint inference do not strongly changes"*
Changed.

*Page 17 line 33: Change the model "wants to go back to" to "returns"*
Changed, see reviewer 1.

*Page 20 line 9: Replace "more at a loss" with "negative".*
Updated.

*Page 20 line 10, Simplify the sentence starting with "If we now...", to read "The CR rates are significantly faster than the AR ensemble rates, and the slopes of the rates (i.e. the second derivative of the concentration evolution) also differ".*
Done.

Figure 9: These are difficult to compare with the different y axis. Consider an additional plot that contains net production and net loss for each run, and including these all on the same axis (without individual reactions).
We have added an additional plot that only contains the net fluxes for each run:

---

## Author Comment (AC5) · 19 Dec 2017

**Correction of our response to reviewer #2**

*Anonymous Referee #2*

*[...]*
*Figure 9: These are difficult to compare with the different y axis. Consider an additional plot that contains net production and net loss for each run, and including these all on the same axis (without individual reactions).*

Unfortunately, when made a mistake when submitting our replies to reviewer 2 and included an erroneous revised version of Figure 9. Please find the correct new version of Figure 9 here below:

[Figure]